# Natriuretic Peptides, Cognitive Impairment and Dementia: An Intriguing Pathogenic Link with Implications in Hypertension

**DOI:** 10.3390/jcm9072265

**Published:** 2020-07-16

**Authors:** Giovanna Gallo, Franca Bianchi, Maria Cotugno, Massimo Volpe, Speranza Rubattu

**Affiliations:** 1Department of Clinical and Molecular Medicine, School of Medicine and Psychology, Sapienza University of Rome, 00189 Rome, Italy; giovanna.gallo@uniroma1.it (G.G.); massimo.volpe@uniroma1.it (M.V.); 2IRCCS Neuromed, 86077 Pozzilli (Isernia), Italy; franca.bianchi@neuromed.it (F.B.); maria.cotugno@neuromed.it (M.C.)

**Keywords:** natriuretic peptides, brain, cognitive decline, dementia, cardiovascular diseases, stroke, hypertension

## Abstract

The natriuretic peptides (NPs) belong to a family of cardiac hormones that exert relevant protective functions within the cardiovascular system. An increase of both brain and atrial natriuretic peptide levels, particularly of the amino-terminal peptides (NT-proBNP and NT-proANP), represents a marker of cardiovascular damage. A link between increased NP levels and cognitive decline and dementia has been reported in several human studies performed both in general populations and in cohorts of patients affected by cardiovascular diseases (CVDs). In particular, it was reported that the elevation of NP levels in dementia can be both dependent and independent from CVD risk factors. In the first case, it may be expected that, by counteracting early on the cardiovascular risk factor load and the pathological processes leading to increased aminoterminal natriuretic peptide (NT-proNP) level, the risk of dementia could be significantly reduced. In case of a link independent from CVD risk factors, an increased NP level should be considered as a direct marker of neuronal damage. In the context of hypertension, elevated NT-proBNP and mid-regional (MR)-proANP levels behave as markers of brain microcirculatory damage and dysfunction. The available evidence suggests that they could help in identifying those subjects who would benefit most from a timely antihypertensive therapy.

## 1. Introduction

Natriuretic peptides (NPs) are a family of cardiovascular hormones mainly secreted by the heart (atrial (ANP) and brain (BNP) natriuretic peptides) and by the endothelium (C-type natriuretic peptide (CNP)) that play important protective functions within the cardiovascular system [1,2]. Apart from their relevant implications in pathophysiology, diagnosis, prognosis and therapeutics of cardiovascular diseases (CVD) [3], a role of BNP and ANP has been convincingly documented in cardiovascular prevention in several population studies, both in apparently healthy individuals and in CVD affected patients [4]. In all circumstances, higher levels of the amino-terminal natriuretic peptides (NT-proBNP and NT-proANP), the more stable forms, predict future cardiovascular events [4]. Whereas the prognostic impact of NPs in patients with CVD can be easily explained as a reflection of the underlying cardiovascular dysfunction and damage, the predictive role of increased NP levels toward future cardiovascular events in apparently healthy individuals is a very intriguing and still unexplained issue. Based on current knowledge on the functional role of the system, it may be supposed that an elevated level of the amino-terminal-NPs in apparently healthy subjects is an index of a subtle initial cardiac and vascular damage that becomes later an overt CVD condition. Therefore, NPs appear able to detect the cardiovascular damage earlier before it could be clinically diagnosed and, while serving as useful markers, they play an important role in the activation of a timely defensive reaction. However, despite these important observations, no clinical guidelines include the use of NT-proBNP level for CVD risk prediction.

Interestingly, the evidence regarding the link between NP level and cardiovascular risk prediction in both healthy subjects and patients with a known history of CVD appears tightly connected to another relevant issue in the context of disease prevention, which is the emerging relationship of NP circulating levels with cognitive decline, vascular dementia and any type of dementia. This important aspect of the pathophysiological implications of NPs has been highlighted over the last recent years, and it certainly deserves to be further characterized. Moreover, its underlying pathophysiological mechanisms need to be better understood.

This article discusses the available evidence on the intriguing relationship between NPs and cognitive decline/dementia, the most plausible explanations and the clinical implications, particularly focusing on hypertension-mediated organ damage.

## 2. Populations-Based Evidence

In the last few years, an increasing body of literature has reported a significant association between NP level and the development of dementia. One of the first investigation on this matter showed that NT-proBNP level predicted accelerated cognitive and functional decline as well as cardiovascular morbidity and mortality in a prospective cohort study of individuals aged 85 years with a 5-year follow-up [5]. In a Japanese population, NT-proBNP was revealed as a biomarker for the future development of dementia [6]. Moreover, some population studies conducted in the general population reported that serum NT-proBNP level was associated with cognitive impairment and microstructural changes detected by neuroimaging. In particular, Zonnefeld et al. [7] detected an association between NT-proBNP level and reduction of total brain volume, of grey matter volume and of microstructural organization of normal white matter, with a consequent increase of white matter volume. In a community-based middle-aged cohort, higher levels of NT-proBNP were significantly associated with a smaller total grey matter volume, although this association was attenuated after adjusting for cardiovascular risk factors and cardiac output [8]. Thus, based on these observations and on the well-recognized heart–brain link [9], NT-proBNP level can be interpreted as a marker of subclinical cardiac dysfunction, underlying subclinical brain damage and ultimately dementia. If this hypothesis holds true, it is expected that, by disrupting early the pathological processes leading to a rise of NT-proBNP level, we can prevent dementia in older people.

A recent study underscored for the first time the relevance of incremental changes of NT-proBNP level over time by reporting that they were able to predict future dementia in a Caucasian population [10]. In this study, the baseline NT-proBNP level was associated with the future development of impaired cognitive function. Most importantly, a 3-year increase of NT-proBNP level over time was associated with an increased risk of future dementia whereas a decrease of NT-proBNP level was associated with reduced risk of dementia. The NT-proBNP level increase was correlated to the presence of cardiovascular risk factors and concomitant comorbidities, such as impaired renal function, hypertension, diabetes mellitus, smoking habit and coronary artery calcification, which represent potential targets in order to prevent dementia [11,12]. In fact, this study suggested that early treatment of these risk conditions and adequate cardiovascular prevention could avoid dementia development in the elderly. Notably, a higher circulating BNP level appears as a suitable marker for adequate interventional strategies toward these risk conditions and for a successful prevention of progressive cognitive impairment and dementia.

In line with this evidence, a study by Hilal et al. showed that a significant association of NT-proBNP and cognitive impairment exists only in the presence of cerebrovascular disease. Herein, NT-proBNP should be considered as a marker of ischemic brain damage [13,14]. These studies supported the role of circulating markers of cardiac dysfunction reflecting silent brain injury or systemic vascular damage.

On the other hand, few studies have shown that the elevation of NPs in dementia can also occur independently from CVD risk factors and that an increased NT-proBNP level is an independent risk marker for dementia particularly among men [15]. Thus, based on this evidence, it has been proposed that BNP may be a direct marker of neuronal damage and of a pathogenic process located within the brain. In accordance with this hypothesis, the plasma level of BNP was found associated with levels of amyloid-β in the cerebrospinal fluid [16]. The association of NT-proBNP level with dementia in a CVD-free population, as reported by Tynkkynen et al. [15], indicates that the neurodegenerative changes start very early in the course of CVD, as reflected by early changes of the NT-proBNP level. The data obtained by Ferguson et al. in a middle-aged population also support, at least in part, a direct link between BNP and brain damage [8]. Accordingly, in a study by Sabayan et al. [17], a higher NT-proBNP level was associated with alterations of brain structure and function, independently of cardiovascular risk factors and of cardiac output, suggesting that NT-proBNP may be directly related to age-dependent structural and functional brain changes, including decline in brain tissue volume, cognitive impairment, and increased depressive symptoms. Thus, BNP level can be considered as a potential marker for timely preventive and treatment strategies in order to avoid development of dementia later in life [18].

With regard to the relation with age, it is evident that the majority of the studies showing an association between elevated BNP level and mild cognitive impairment (MCI), including conversion from MCI to Alzheimer’s disease (AD), and with AD and vascular dementia are confounded by the fact that, often, the patient population is older than the controls. However, few studies have shown that the increase of BNP may be also independent from age. In particular, a study by Hiltunen et al. demonstrated that its prognostic capacity may be only valid among subjects below 79 years of age [19]. With regard to sex, women are disproportionately affected by AD and other types of dementia compared to men, have a greater risk of developing cognitive decline in the presence of risk factors and experience a faster progression of hippocampal atrophy [20,21]. However, although NP levels are higher in women [22], no evidence exists about its influence in the different evolutions of dementia according to sex.

Another important component of the NP family was explored for its ability to predict and possibly associate with dementia. In fact, it was demonstrated that the mid-regional (MR)-proANP level behaved as a marker of microvascular dysfunction and neurodegenerative process in the transition from MCI to AD. Interestingly, MR-proANP played its predictive role independently from blood pressure (BP) level [23]. More recently, an elevated level of MR-proANP was reported to be independently associated with a higher risk of incident all-cause and vascular dementia in a population-based prospective study [24]. Thus, these studies highlighted a CVD-independent role of ANP in dementia and supported some of the findings previously obtained with regard to a direct role of BNP in the brain [15,18].

## 3. Mechanistic Insights on the Link between NPs and Cognitive Decline/Dementia

### 3.1. CVD-Dependent Pathogenic Mechanisms

As discussed above, a high NT-proBNP level associates closely with white matter microstructural damage and brain atrophy in subjects with prior onset of CVD and diabetes and in subjects without cardio- and cerebrovascular diseases. Due to the fact that the link between NPs and risk of dementia can be both dependent and independent from CVD and cardiovascular risk factors, the exact pathophysiological mechanisms underlying this association are complex and they are still in part unclear. It may be supposed that subjects with subclinical vascular disease, reflected by an increased NT-proBNP level, first manifest CVD and then dementia. Since BNP is protective and not neurotoxic, its increase should be mainly considered as a marker reflecting mechanisms of cerebral hypoperfusion and neurodegeneration, microemboli and cardioembolic stroke, as the consequence of the cardiovascular risk factors load [25]. In fact, the latter may contribute to the development of atherosclerotic disease responsible for ischemia, endothelial dysfunction, abnormalities in microcirculation, hypoxia, increased inflammation and oxidative stress, interstitial fibrosis, breakdown of blood–brain barrier (BBB) function and damage of the neurovascular unit. The establishment of cerebral microinfarcts might explain the consequent development of clinical dementia. For instance, heart failure leads, as a consequence of reduced cardiac function, to cerebral hypoperfusion, oxidative stress and neuronal dysfunction via deposition of amyloid-β or neurovascular damage [26]. In hypertension, the loss of cerebral vascular autoregulation, resulting in hypoxia and regional cerebral atrophy, mediates the reduction of cognitive function. Thus, an increased NT-proBNP level reflects subclinical disease, particularly poor cardiac function and volume overload in the presence of CVD. In this view, the BNP level increase may reflect a combination of cardiac, neurovascular and neurodegenerative aetiologies. It is also possible that common systemic vascular processes drive both cardiac and brain pathologies. Importantly, given that a higher NT-proBNP level is detected in subjects with higher loads of vascular damage, it should be assessed if timely treatment of CVDs and of cardiovascular risk factors would influence the link between NT-proBNP and brain structural and functional impairments, ultimately preventing the development of cognitive impairment and dementia.

### 3.2. CVD-Independent Pathogenic Mechanisms

As mentioned before, a second potential mechanism linking NPs and cognitive impairment/dementia is a direct relationship between NT-proBNP and brain structure explained by the presence of BNP [27] and of its receptor in neuronal tissue [28,29]. Several animal and human studies showed an abundant extensive distribution of all NPs in the central nervous system [27]. Their receptors may affect cerebral function through the regulation of BBB integrity, synaptic transmission, and modulation of both systemic and central nervous system stress responses. In fact, it is known that NPs are involved in neural development, in neurotransmitter release, in synaptic transmission, in regulation of inflammatory processes at the level of microglia and in neuroprotection [27,28,29,30]. Consistently, ANP has been shown as a neuroprotective agent via upregulation of the Wnt/β-catenin pathway in an in vitro model of Parkinson disease [31]. An increase of ANP upon neprilysin inhibition therapy protected from stroke occurrence in a high-salt-fed stroke-prone spontaneously hypertensive rat model [32]. Furthermore, an increase of circulating ANP levels predicted stroke in apparently healthy individuals [33]. Based on this knowledge, it may be expected that increased levels of NP can indicate the presence of initial brain and vascular damage that will become later on a conclamant disease condition. In this context, as pointed out in the introduction of this article, the NP increase could be considered both as a marker and as a defensive endogenous reaction toward brain damage. Moreover, due to the role that NPs play in regulating several functions that are involved in the course of cognitive impairment, we cannot definitively rule out the possibility that abnormalities of NP regulation and function, as expressed by high levels, may themselves be involved in the development of cognitive impairment/dementia. In all cases, the measurement of NT-proNP levels may represent a useful tool to identify individuals at high risk of developing abnormal brain aging.

It is interesting to note that the protective role of NPs in the brain is strongly supported by the positive results obtained in the past with the use of nesiritide (a human recombinant form of BNP). In fact, its infusion led to an improvement of cerebral blood flow and of brain injury, with a better functional outcome [34]. Moreover, infusion of BNP in ischemic animal models showed a significant infarct volume reduction and a better sensorimotor recovery [35]. The same study reported that patients affected by cardioembolic stroke carrying higher circulating BNP levels presented a better outcome at a 3-month follow-up. Moreover, available evidence shows that low concentrations of NPs in the brain along with elevated systemic concentration are linked to structural and functional cerebral alterations. On the other hand, higher levels of NPs in the brain are crucial for the maintenance of brain homeostasis [27]. This evidence supports the hypothesis that BNP exerts direct protective effects in the brain through its own receptor, its increase may truly be a marker of an ongoing pathological process within the brain and, finally, elevation of BNP level may be a suitable target for therapeutic purposes.

It is likely that both direct and indirect mechanisms underlie the pathogenic relationship between NPs and cognitive decline/dementia. We also need to point out that the intriguing relationship does not necessarily mean causation but that it may rather indicate a correlation/association between NPs and cognitive deficit/dementia.

## 4. NPs, Cognitive Impairment and Dementia: Implications in Hypertension

Hypertension represents one of the most commonly known risk factors for cognitive impairment and dementia, also including low levels of education, smoking, high total cholesterol, obesity, diabetes, atherosclerosis, arteriolosclerosis, hypertensive vascular damage, impaired vascular autoregulation, heart failure and anaemia. Previous studies showed that high-to-normal BP levels correlated with lower grey matter volume in several brain regions in young adults. Thus, BP-associated grey matter alterations start early in adulthood and emerge continuously across the range of BP [36]. High BP in mid-life has been involved in the development of dementia [37] and even with AD-type pathophysiology, later in life [38]. Interestingly, BP level is increased from 5 to 15 years before the onset of dementia, and it declines within the years before overt development of dementia [39].

The relationship between BP and cognitive decline is known to be age dependent. In particular, low BP level has been constantly associated with poor cognitive function especially in older subjects. It is known that autoregulation of cerebral vasculature increases the blood flow to the brain during reduced cardiac function and in the presence of low BP levels. However, this compensatory mechanism is compromised when systemic blood flow reduction is chronic or subclinical, particularly in the elderly. On the other hand, since low BP levels associate with impaired cognitive function in the elderly, it is likely that elevated BP in the same range of age is an indicator of good cardiac pump function required for adequate perfusion of the brain.

Interestingly, it was observed that the harmful effect of hypertension on the risk of dementia is higher in females than in males [40]. Also, blacks are more likely than whites to develop hypertension and the related cognitive decline [41].

At the tissue level, hypertension induces alterations in neurovascular function that, together with reduced cerebral perfusion, alterations in BBB permeability and deficiency of vascular growth factors, alter neuronal function in regions involved in cognitive function (hippocampus, entorhinal cortex and prefrontal cortex) [42]. At the molecular level, the processes of inflammation and of oxidative stress within the brain tissue may contribute to the link between subclinical CVD and neurodegenerative changes in hypertension (Figure 1), particularly in the presence of other vascular risk factors such as diabetes. In fact, endothelial dysfunction may also play a role. In this context, biomarkers for the microvascular contribution to cognitive impairment and dementia in hypertension are needed.

The relevance of the NP system in this regard is supported by evidence linking BNP to arterial stiffness and BP variability. Arterial stiffness is known to predict cognitive decline in hypertension [43,44,45]. Of interest, a recent study found a significant association between higher NT-proBNP level and increased arterial stiffness modulated by BP variability [46]. This finding provides a major strength to the pathophysiological implications of BNP, as a marker of arterial stiffness, in the prediction of cognitive decline/dementia in hypertension.

It has been reported that NT-proBNP level reflects poor cardiac function and volume overload in hypertension [47]. An increased NT-proBNP level was independently associated with the presence of subclinical MRI signs of brain small vessel disease in a cohort of hypertensive patients free of stroke and dementia [48]. Thus, NT-proBNP level may provide a useful inexpensive tool to identify subclinical CVD causing subclinical neurodegenerative changes and the need for a timely intervention strategy. In this regard, a few limitations need to be taken into account when using NPs as markers. For instance, it is still uncertain the role that NPs may play in the higher predisposition of both female sex and black subjects to develop cognitive decline and dementia in the presence of hypertension. Of note, females have higher NP levels [22] and blacks have lower NP levels [49]. Apart from sex and race, NP levels are known to be influenced by age, renal function, body mass index and comorbidities so that the assessed plasma NP level has to be considered once adjusted for the above parameters [4].

A study by Kerola et al. [50] reported a trend toward low diastolic BP and new onset of dementia. In this study, the diagnosis of hypertension was associated with lower incidence of dementia in the follow-up, independent of known risk factors for dementia, and it was a likely consequence of the appropriate antihypertensive medications used. Notably, this study, while demonstrating an independent predictive role of BNP towards dementia in the elderly, indicated that cardiovascular morbidity and stress significantly affected cognitive decline in older subjects. Therefore, early initiation of antihypertensive therapy appears of crucial relevance for the prevention of dementia development. In this regard, a few trials have also shown that treatment of hypertension lowered the incidence of cognitive decline in the elderly [51,52,53]. Thus, it is worthwhile considering the impact of antihypertensive medications on cognitive decline in this group of subjects based on the level of BNP, and, particularly, considering the opportunity to start early the treatment in those patients with elevated BNP levels.

Of note, antihypertensive therapy was shown to reduce the rate of conversion from MCI to AD also in subjects with high MR-proANP [54]. Based on these findings, elevated MR-proANP, as a marker of microcirculatory function in the brain, could help to identify those subjects who would most benefit from antihypertensive therapy.

However, contrasting data also exist. Some studies have demonstrated that treatment of hypertension decreased the risk of vascular and all-cause dementia but it did not decrease the risk for AD, cognitive impairment and cognitive decline. Of note, there are also data regarding the effect of specific antihypertensive treatments. Observational studies reported potential preventive effects toward cognitive decline and dementia with the use of calcium channel blockers and of renin-angiotensin system blockers [55]. Another study [56] has reported that the use of angiotensin converting enzyme inhibitors (ACEI) might associate with a better outcome in terms of cognitive decline. This is probably secondary to the decrease in cardiac workload with a consequent reduction of BNP level rather than to a direct effect of ACEI on local BNP in the brain. The beneficial effects of ACEI toward cognitive impairment were more recently confirmed [57]. In the Perindopril protection against recurrent stroke study (PROGRESS), use of perindopril or indapamide led to a benefit in dementia reduction among individuals with recurrent stroke [58]. In a recent study, telmisartan (an AT1R blocker) synergistically interacted with low-dose rosuvastatin to reduce white matter hypertensive progression and cognitive function decline [59]. Contrasting findings are available with the new pharmacological approach based on neprilysin inhibition in association with an AT1R blocker (ARNi) [60]. Since neprilysin normally degrades amyloid-β, its inhibition was initially seen with particular concern due to the possible risk of increased occurrence of dementia and AD [61]. Other studies, however, have denied a relationship between neprilysin inhibition and AD. It has been supposed that the increased NPs level, as a consequence of ARNi treatment, may counterbalance the detrimental effects of cerebral deposition of amyloid-β [62,63,64].

It is clear that the optimal antihypertensive regimen able to prevent future cognitive decline or dementia is difficult to assess for each individual, considering that comorbidities, socioeconomic and demographic characteristics may impact on the effect of the therapy.

Based on all abovementioned evidence, it is recommended to diagnose early high BP and to start an appropriate treatment with the aim to avoid cognitive decline and dementia in elderly subjects [65,66]. In this regard, higher NT-proBNP and MR-proANP levels may adequately identify subgroups of patients with marked endothelial dysfunction and microvascular pathology as the most suitable inexpensive markers to indicate a timely and efficacious therapeutic intervention to prevent dementia in hypertensive patients. Finally, based on current knowledge on the neurological functions of NPs, these hormones and their receptors might be a suitable therapeutic target to treat cognitive impairment and dementia (Figure 1).

## Figures and Tables

**Figure 1 jcm-09-02265-f001:**
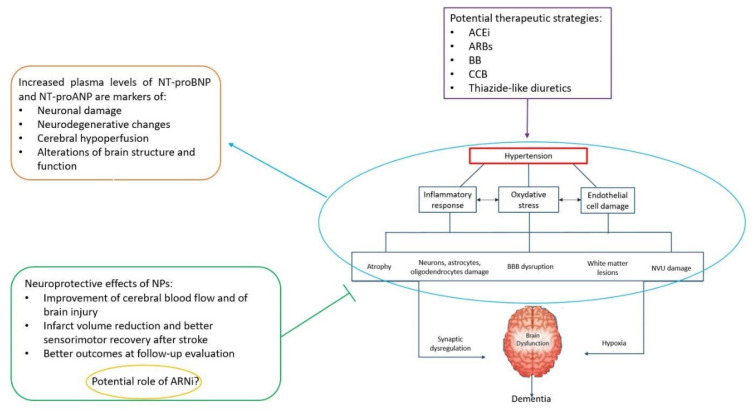
Schematic representation of the pathophysiological mechanisms underlying the development of dementia in hypertension: The role of aminoterminal natriuretic peptides (NT-proNPs) as both markers of brain damage and neuroprotective agents is highlighted. An appropriate antihypertensive treatment can be started on time by taking advantage of the NT-proNP levels in order to counteract the evolution to dementia in older patients.

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
