# Peer review of "Natriuretic Peptides, Cognitive Impairment and Dementia: An Intriguing Pathogenic Link with Implications in Hypertension"

_jcm, 2020, doi:10.3390/jcm9072265_

Round 1

Reviewer 1 Report

This article discusses the available evidence on the relationship between NPs (natriuretic peptides) and cognitive decline / dementia, the most plausible explanations and  the clinical implications. This review discusses the implications hypertension mediated organ damage.

The increase of NPs level in dementia can be both dependent and independent from CVD risk factors. In case of a link independent with the CVD risk factor, an elevated NPs level should be considered a direct marker of neuronal damage. In the context of hypertension, elevated NT-proBNP and mid-regional (MR) -proANP levels behave as markers of brain microcirculatory damage and dysfunction.

The review is well written and easy to understand.

Here are my specific comments:

- There is evidence in the literature that BP levels are related to both age and sex. Are there articles showing relationships between sex, BP levels and cognitive decline?

- Many studies show an association between high BNP level and Alzheimer's disease (AD). Are there correlations also with other neurodegenerative diseases?

Author Response

We wish to thank this Reviewer for his/her positive comments on our article. We took in great consideration the specific comments provided in order to improve the manuscript presentation.

Our replies follow:

  1. Based on recent reports, female sex has a higher risk to develop dementia in the presence of hypertension. This has been added in the revised text (pg. 3, lines 127-128; new refs. 20, 21).
  2. Most of the correlations have been described for BNP/ANP and AD. Few studies explored the role of NPs in Parkinson disease (PD), mostly exploring the cardiac implications of elevated NPs level in this disease. We have already mentioned the paper by Colini Baldeschi reporting an important neuroprotective effect of ANP in an in vitro model of PD (ref. 31 of the revised ms).

Reviewer 2 Report

The manuscript by Gallo et al summarizes and discusses the role of natriuretic peptides as potential markers of cognitive decline in hypertension. The paper is very well written and is a pleasure to read. I have a couple of suggestions for the authors to improve the manuscript. 

  1. Please make sure to emphasize and discuss, when relevant, that correlation/association does not equal causation, and emphasize the limitations of using NPs as markers in a separate subsection. 
  2. The authors mention (lines 106-108) some potential sex differences in NP levels. Are there any associations in dementia between males and females, and could this be related to NP levels? 
  3. When talking about mechanisms (part 3) please separate possible mechanisms into subsections with titles
  4. Some populations have counterintuitively lower levels of ANP and higher BP (such as salt-sensitive Black cohort). Is there data to correlate these findings with dementia and cognitive decline? 
  5. The authors mention ARNi; is there data about the effects of Entresto on cognitive function? 
  6. Some minor proofreading is needed (like grammar in line 209)

Author Response

We wish to thank very much this Reviewer for his/her positive comments on our article. We took in great consideration the specific comments provided in order to improve the manuscript presentation.

Below are our replies:

  1. We followed the Reviewer’s suggestion and added specific sentences related to her/his concerns in the revised text (see pg. 5, lines 206-208; pg. 6, lines 250-256; new refs. 22, 49).
  2. As suggested also by Reviewer 1, we added the evidence regarding difference in dementia occurrence between males and females in the revised ms (see pg. 3, lines 127-128; new refs. 20, 21). We also commented about the lack of evidence of a role of NPs in the difference between sexes (see pg. 6, lines 250-253; new refs. 22, 49).
  3. We followed the Reviewer’s suggestion and divided the paragraph 3 into subsections.
  4. This is a very interesting matter that was introduced in the revised text (pg. 5, lines 228,229, new ref. 41; pg. 6, line 253; new refs. 22, 49). However, as commented on pg. 6, lines 251-253, there is no evidence of a role of NPs in the higher predisposition to cognitive decline of hypertensive black subjects.
  5. With regard to ARNi, we had previously reported in the paragraph 4 (NPs, cognitive impairment and dementia: implications in hypertension), the available evidence on the association of this new class of drug with cognitive decline-dementia. A new literature search did not reveal anything new compared to what already staten in our previous version of the article (current pg. 6, lines 286-292 of the revised ms, refs of the revised ms. 61-64).
  6. Thank you for the comment on the presence of few grammar errors in the text. We checked the ms to correct them.
